# PGMPL: PROTOTYPE-GUIDED MULTI-MODAL PROMPT LEARNING FOR VISION-LANGUAGE MODELS

## ABSTRACT

Vision-language models (VLMs) have been widely applied to various visual tasks due to their strong zero-shot transfer capabilities. However, their performance on downstream tasks often remains suboptimal. While fine-tuning can improve accuracy on base classes, it often compromises generalization to novel classes. To address this challenge, we propose the **P**rototype-**G**uided **M**ulti-modal **P**rompt **L**earning (PGMPL), which guides representation learning through a supervisory signal with intra-class summary information. Specifically, we construct a category-level prototype for each class by aggregating multi-image features with textual semantics. This prototype serves as a cross-modal, summarizing supervisory signal, strengthening image-text alignment and enhancing the generalization of the learned representations. To further optimize prototype and its guidance of representation learning, we refine multi-modal representations via prompt learning and introduce bidirectional cross-attention to alleviate the image-text matching inconsistency induced by newly inserted prompts. Extensive experiments demonstrate the effectiveness of PGMPL, which achieves a higher overall harmonic mean than state-of-the-art methods in zero-shot tasks across 11 datasets. Our code is available at `https://anonymous.4open.science/r/PGMPL`.

## 1 INTRODUCTION

In recent years, vision-language models (VLMs) such as CLIP (Radford et al., 2021) have been widely applied to various open-world vision tasks due to their strong zero-shot transfer capabilities. Trained on large-scale image-text pairs, VLMs establish well-aligned embedding spaces across modalities. However, they face critical challenges in adapting to downstream tasks through fine-tuning while preserving the generalization capabilities. For instance, in few-shot fine-grained retrieval, it can fine-tune the model on several images of Siamese cat to improve retrieval of Siamese cats; yet when confronted with unseen cat breeds such as Bengal, the model fails to discriminate them, revealing overfitting and insufficiently generalizable features learned from limited samples.

Prompt learning is a lightweight approach to improve model's representations by inserting learnable tokens into text or image embeddings. For example, CoOp (Zhou et al., 2022c) and CoCoOp (Zhou et al., 2022b) insert learnable tokens into text embeddings while freezing the pretrained CLIP model, and MaPLe (Khattak et al., 2023) and MMRL (Guo & Gu, 2025a) extends this idea to images, establishing cross-modal mappings between image and text to enhance representation quality. However, existing methods that focus mainly on text-image contrastive learning tend to overlook the learning of generalizable visual and textual representations, leading to limited generalization in new scenarios. Therefore, it is essential to develop an image encoder that can simultaneously ensure strong image-text alignment and maintain high generalization capability for its visual representations.

To address the above issues, we propose the **P**rototype-**G**uided **M**ulti-modal **P**rompt **L**earning (PGMPL). Reflecting on human concept formation, a single image is often insufficient to reveal the essential traits that define a "Siamese cat". By observing multiple instances, we abstract the stable, class-specific attributes, such as the short hair and blue eyes, while suppressing incidental background interference. Guided by this insight, we introduce the concept of a category-level prototype, a summarizing supervisory signal designed to mimic this human-like abstraction process by aggregating multiple images. It simultaneously maintains image-text alignment and guides the learning of more generalizable visual representations. Specifically, during training, we construct and maintain

a prototype for each category by using cross-attention to fuse image and text features, yielding a stronger and modality-bridging supervisory signal to guide representation learning. Better vision-language representations lead to more reliable prototypes, which in turn provides stronger guidance for representation learning. To avoid the generalization drop during fine-tuning both image and text encoders, we adopt parameter-efficient prompt learning to optimize the features. Furthermore, we introduce an image-text interaction mechanism, to prevent prompt introduction from disrupting image-text matching consistency. This mechanism is a bidirectional cross-attention interaction method based on batch tokens, enabling aligned information exchange between the two modalities within the intermediate layers of encoders, thereby preserving image-text matching consistency.

We conduct extensive experiments under various settings, including base-to-novel, cross-dataset, and cross-domain image-text matching, as well as image-cluster feature matching generalization. Results on 11 datasets show that our method improves novel-class generalization by 0.45% and average performance by 0.33% compared to state-of-the-art methods, while maintaining base-class accuracy. Additionally, our accuracy on cross-dataset and cross-domain tasks remains comparable to current state-of-the-art approaches. Under the image-cluster feature matching setting, our method outperforms state-of-the-art methods on both base and novel classes, achieving an average improvement of 1.75%, which demonstrates stronger vision feature representation capabilities. In summary, our contributions are as follows:

(1) We propose a prototype-guided multi-modal prompt learning method PGMPL, which utilizes a prototype with category-level summarizing information as novel supervisory signals to enforce discriminative representation learning across seen (base) and unseen (novel) classes, significantly boosting CLIP's generalization ability.

(2) We introduce batch tokens with bidirectional cross-attention interaction mechanism to optimize representations and enable aligned information exchange between the image and text encoder, thereby maintaining consistent image-text matching.

(3) Extensive experiments show that our method outperforms state-of-the-art methods across various settings on 11 datasets, which demonstrates its superior generalization and feature representation capabilities on both base and novel classes.

## 2 RELATED WORK

### 2.1 VISION-LANGUAGE MODELS

Vision-Language Models (VLMs), exemplified by architectures like CLIP (Radford et al., 2021), FILIP (Yao et al., 2021), ALIGN (Jia et al., 2021), LiT (Zhai et al., 2022), VILA (Lin et al., 2024), and SigLIP (Zhai et al., 2023; Tschannen et al., 2025), establish cross-modally aligned joint embedding spaces through contrastive learning on large-scale image-text datasets, demonstrating robust zero-shot generalization performance. Due to its powerful image and text representation capabilities, VLMs have been widely used in various downstream tasks, such as dense prediction (Rao et al., 2022; Zhou et al., 2022a), action understanding (Nichol et al., 2021; Ramesh et al., 2022; Patashnik et al., 2021), image and video captioning (Barraco et al., 2022; Mokady et al., 2021; Tang et al., 2021), and visual question answering (Wang et al., 2023; Özdemir & Akagündüz, 2024). However, the massive training data requirements brings high computational costs, making task-specific fine-tuning particularly resource-intensive. Critically, parameter updates of VLMs may lead to overfitting and, consequently, degrade their generalization capabilities. Therefore, how to adapt VLMs more efficiently to specific downstream tasks remains a critical challenge.

### 2.2 PROMPT LEARNING

To enhance generalization while avoiding high computational costs and performance degradation caused by fine-tuning, contemporary approaches employ prompt learning to optimize VLMs by inserting learnable tokens into text or image embeddings. CoOp (Zhou et al., 2022c) utilizes learnable text prompts to replace hand-crafted templates, inserting trainable tokens into text embeddings. Optimized through limited image-text pairs, it significantly enhances CLIP's performance on base classes. To further strengthen generalization to novel classes, CoCoOp (Zhou et al., 2022b) introduces an image-conditioned dynamic prompt framework that mitigates overfitting risks in base class

tasks. KgCoOp (Yao et al., 2023) employs knowledge regularization, constraining learnable prompts using frozen CLIP's hand-crafted prompt features to balance performance in base and novel class. Beyond text prompt learning, MaPLe (Khattak et al., 2023) introduces learnable visual tokens in image encoders, establishing cross-modal mapping with textual tokens for joint vision-language representation optimization. ProVP (Xu et al., 2025) advances this through progressive visual prompts that enhance inter-layer interaction, ensuring deep propagation of visual embeddings. ATPrompt (Li et al., 2024) innovates with a novel prompt paradigm, injecting multiple universal attribute tokens into learnable soft prompts to strengthen alignment between image features and unknown categories. MMRL (Guo & Gu, 2025a) and MMRL++ (Guo & Gu, 2025b) construct shared multi-modal spaces, projecting learnable space tokens into textual and visual representation spaces to facilitate cross-modal interaction.

However, the over-emphasis on direct image-text alignment in existing methods often leads to the neglect of learning representations within each modality, thereby limiting generalization. To address this, we propose PGMPL, which constructs a modality-bridging prototype for each category. This prototype acts as a superior supervisory signal that both ensures image-text alignment and guides the model to learn more generalizable, class-discriminative representations.

## 3 METHOD

In this section, we first review VLMs (*e.g.*, CLIP) and prompt learning techniques for improving CLIP's generalization. Then we detail the core components of our PGMPL, including learnable batch tokens with bidirectional cross-attention interaction and prototype-guided prompt learning.

### 3.1 PRELIMINARIES

Consistent with prior work, we adopt CLIP (Radford et al., 2021) as VLM, which comprises an image encoder $\mathcal{V}$ and a text encoder $\mathcal{T}$. Given an image embedding $v_i$ and $N$ class-specific text embeddings $\{t_j\}_{j=1}^N$, the prediction results of CLIP are as follows:

$$p(y|v_i) = \frac{\exp\left(sim\left(\mathcal{V}(v_i), \mathcal{T}(t_y)\right)/\tau\right)}{\sum_{j=1}^N \exp\left(sim\left(\mathcal{V}(v_i), \mathcal{T}(t_j)\right)/\tau\right)}, \tag{1}$$

where $sim(\cdot, \cdot)$ denotes inner product, and $\tau$ is a temperature parameter.

Despite CLIP's strong zero-shot performance, fine-tuning is necessary for specific tasks. Fine-tuning the encoders can degrade the model's generalization capability. To address this, prompt learning is proposed as an effective technique for enhancing the performance of CLIP. Specifically, it augments representations by inserting learnable tokens into image and text embeddings. Given a template `"a photo of a [CLS]"`, tokenized as $t = [t_1, t_2, \ldots, t_X]$, $Y$ learnable tokens are added to form text embedding $t' = [t_1, \ldots, t_X, c_1, \ldots, c_Y]$. Similarly, we can obtain the image embedding $v'$. The enhanced features $\mathcal{T}(t')$ and $\mathcal{V}(v')$ then replace their original counterparts in Eq. (1).

### 3.2 MOTIVATION

Existing methods focus mainly on text-image contrastive learning, which overlooks the learning of generalizable visual and textual representations, thereby limiting their generalization to new scenarios. A more representative, modality-bridging supervisory signal maybe the solution. Therefore, starting from intra-class clustering representations, we explore shared features of a category, with the expectation that the model can learn the essential distinctions cross classes. This aspect has been

Table 1: CLIP's accuracy on base and novel classes across 11 datasets. $n$ is the number of images used to form the cluster feature.

| $n$ | Base | Novel | HM |
|---|---|---|---|
| 1 | 50.30 | 53.96 | 52.07 |
| 5 | 65.70 | 68.38 | 67.01 |

overlooked in previous research. The results in Table 1 support our idea: aggregating multiple images from the same class (*e.g.*, by averaging) to form a category-level cluster feature for classification significantly outperforms using a single image as the cluster feature. This is because multi-image aggregation extracts category-level stable factors, suppressing incidental noise and background in

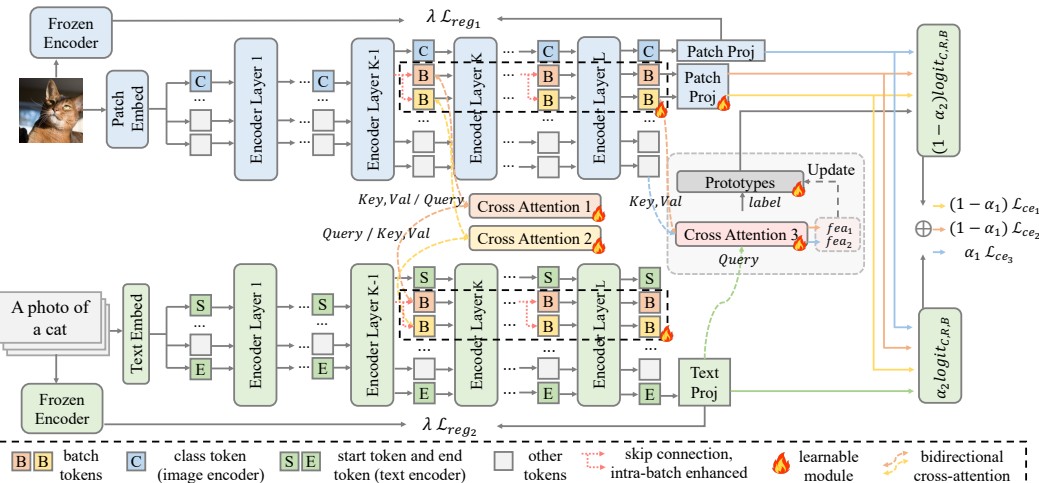

Figure 1: Overview of the proposed PGMPL. PGMPL consists of two main components: the first is the gray area, which involves the construction and maintenance of the prototype. The second component is the introduction of learnable batch tokens with a bidirectional cross-attention interaction mechanism in the intermediate layers of the CLIP encoder.

individual images and yielding a more class-summarized representation. Based on this finding, we introduce a novel supervisory signal (prototype) that acts as a cluster center during training. Guided by this prototype, model can focus on shared class attributes while ignoring instance-specific noise, thereby ensuring image-text alignment and enhancing the generalization of representations.

## 3.3 PGMPL: PROTOTYPE-GUIDED MULTI-MODAL PROMPT LEARNING

To address the key challenges of VLMs in zero-shot tasks, we propose **P**rototype-**G**uided **M**ulti-modal **P**rompt **L**earning (PGMPL), as illustrated in Figure 1. Inspired by how humans form concepts from multiple instances, PGMPL aggregates multi-image features per class together with textual semantics to form a stable, category-level supervisory signal (*i.e.*, prototype) to guide better representation learning. To obtain reliable prototypes, it is necessary to ensure the effectiveness of features extracted by the encoders. Specifically, we insert carefully designed learnable batch tokens into intermediate layers of encoders and perform cross-modal fusion via cross-attention, mitigating image-text mismatch that may arise from introducing new tokens. Then, we use the optimized image patch tokens together with learnable tokens to enhance the text representation, producing an enhanced text that is used to update the category-level prototype. The prototype further guides image-text contrastive learning, improving the model's generalization ability to novel classes.

### 3.3.1 LEARNABLE BATCH TOKENS WITH BIDIRECTIONAL CROSS-ATTENTION INTERACTION

To obtain a better prototype, we optimize the features extracted from both the image and text encoders. We rely on prompt learning to refine the representations instead of fine-tuning, which is known to degrade generalization.

**Learnable batch tokens.** We insert learnable tokens into image and text embeddings to optimize image and text representation. We first initialize $M$ batch tokens $B$, where the first $\frac{M}{2}$ tokens are denoted as $B_{\frac{M}{2}}$ and the last $\frac{M}{2}$ tokens are denoted as $B_M$. For each batch, we enhance $B_M$ with $B_{\frac{M}{2}}$ on both the image and text sides, which introduces a category-agnostic semantic representation that to improve feature expressiveness for unseen classes:

$$B_M^{img} = \beta_1 B_M^{img} + (1 - \beta_1) \cdot \frac{1}{Z} \sum_{i=1}^{Z} B_{\frac{M}{2},i}^{img}, \quad B_M^{txt} = \beta_2 B_M^{txt} + (1 - \beta_2) \cdot \frac{1}{Z} \sum_{j=1}^{Z} B_{\frac{M}{2},j}^{txt}, \quad (2)$$

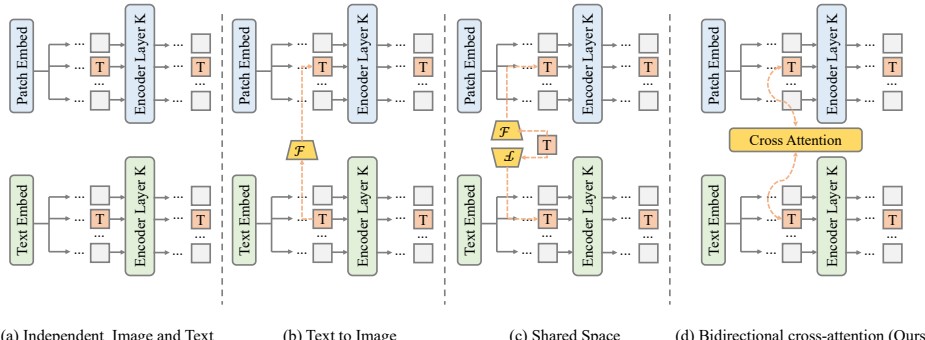

(a) Independent Image and Text     (b) Text to Image     (c) Shared Space     (d) Bidirectional cross-attention (Ours)

Figure 2: Comparison of cross-modal interaction strategies.

where $\beta_1$ and $\beta_2$ are hyperparameters that control the extent to which the tokens utilize batch-wise information, and $Z$ denotes batch size.

Motivated by the observation that features from shallow encoder layers preserve generalizable information, while deeper layers capture specific representations (Yang et al., 2024), we insert tokens starting from an intermediate layer $K$ following Guo & Gu (2025a), to balance the performance on base and novel classes. Furthermore, we implement skip connections between shallow and deep features to explicitly preserve generalizable information:

$$B_{\frac{M}{2},l} = \beta_3 \cdot B_{\frac{M}{2},l} + (1 - \beta_3) \cdot B_{\frac{M}{2},l-\frac{L-K+1}{2}}, \tag{3}$$

where layers $l \in (K, L]$, and $\beta_3$ is hyperparameter that controls the fusion ratio.

**Bidirectional cross-attention interaction mechanism.** Introducing learnable tokens into intermediate encoder layers disrupts the representation distributions of the two modalities and undermines the consistency of image-text matching. Therefore, we incorporate image-text interaction during training to mitigate this mismatch.

Existing methods exhibit significant limitations in cross-modal interaction. As shown in Figure 2, traditional approaches (Figure 2 (a)) lack direct inter-modal interaction, performing only simple matching at the final feature level. Methods like MaPLe (Figure 2 (b)) achieve only unidirectional text-to-image mapping, failing to capture visual feedback for text representations. While shared space methods such as MMRL (Figure 2 (c)) project features from a common representation space to respective modality-specific spaces, their indirect interaction mechanisms result in imprecise semantic alignment and compromised modality-specific information.

The common deficiency across these approaches is the absence of direct pathways between modalities, leading to inadequate cross-modal semantic understanding, particularly for unseen classes. To address this, we construct an bidirectional interaction enhancement module (Figure 2 (d)) that enables direct token-level interaction through bidirectional cross-attention:

$$B^{img} = B^{img} + CrossAttn(B^{img}, B^{txt}), B^{txt} = B^{txt} + CrossAttn(B^{txt}, B^{img}), \tag{4}$$

where $CrossAttn(\cdot, \cdot)$ denotes cross-attention, and the first argument denotes the queries $Q$, the second denotes the keys $K$ and values $V$. Specifically, $CrossAttn(X, Y) = softmax(\frac{Q \cdot K^\top}{\sqrt{d_k}}) \cdot V$, $Q = X \cdot W_Q, K = Y \cdot W_K, V = Y \cdot W_V$, $d_k$ is the dimensionality of $K$.

This symmetric bidirectional architecture preserves modality-specific characteristics while establishing fine-grained cross-modal semantic correlations and enhancing alignment quality of vision-language representations.

Finally, text prompts and visual prompts are formally defined as:

$$[C_l, v_l] = \mathcal{V}_l([C_{l-1}, v_{l-1}]), l = 1, \ldots, K-1, \tag{5}$$

$$[C_l, B^v_{\frac{M}{2},l}, B^v_{M,l}, v_l] = \mathcal{V}_l([C_{l-1}, B^v_{\frac{M}{2},l-1}, B^v_{M,l-1}, v_{l-1}]), l = K, \ldots, L, \tag{6}$$

$$[S_l, t_l, E_l] = \mathcal{T}_l([S_{l-1}, t_{l-1}, E_{l-1}]), l = 1, \ldots, K-1, \tag{7}$$

$$[S_l, B^t_{\frac{M}{2},l}, B^t_{M,l}, t_l, E_l] = \mathcal{T}_l([S_{l-1}, B^t_{\frac{M}{2},l-1}, B^t_{M,l-1}, t_{l-1}, E_{l-1}]), l = K, \ldots, L, \tag{8}$$

where $\mathcal{V}_l$ and $\mathcal{T}_l$ are layer-$l$ operations of image and text encoders. $C_l$ are text class tokens at layer $l$, $S_l$ and $E_l$ are visual start and end tokens at layer $l$, $B_{\frac{M}{2},l}$ and $B_{M,l}$ are batch tokens at layer $l$.

### 3.3.2 PROTOTYPE GUIDANCE

We introduce category-level summarizing prototypes to both strengthen cross-modal alignment and significantly improve inter-class feature discriminability. We use the above-mentioned methods to optimize and extract image and text features. Based on the obtained image and text features, we employ cross-attention that takes the class-specific text feature as the query and the image feature as the key and value to dynamically enhance the text feature. Then, we use the enhanced text feature to update the prototype $p_y$ for each class through a momentum mechanism as follows:

$$p_y = \gamma \cdot p_y + (1 - \gamma) \cdot CrossAttn\left(t_y, P\right) \tag{9}$$

where $t_y$ is the text embedding, $P$ is the patch embeddings of images and batch tokens, and $\gamma$ is a momentum coefficient controlling the update rate of prototypes.

**Training Phrase.** We introduce a dual-objective optimization strategy. Beyond standard vision-language feature alignment, we use prototypes to guide better representation learning, encouraging intra-class compactness and inter-class separation by clustering features around their category-specific prototypes. The logits $s$ can be computed as:

$$s^{img} = \alpha_1 \cdot sim(f_{img}, f_{txt}) + (1 - \alpha_1) \cdot sim(f_{img}, p_y), \tag{10}$$

$$s^{batch} = \alpha_1 \cdot sim(f_{batch}, f_{txt}) + (1 - \alpha_1) \cdot sim(f_{batch}, p_y), \tag{11}$$

where $f_{txt}$ denotes the text feature, $f_{img}$ denotes the image feature, and $f_{batch}$ denotes the batch token feature, including $B_{\frac{M}{2}}$ and $B_M$. $sim(\cdot, \cdot)$ denotes inner product. The corresponding loss is then computed using cross-entropy loss:

$$\mathcal{L}_{CE}^{w} = -\sum_{y \in \mathcal{Y}} y_{true} \log\left(\frac{\exp(s_y^w/\tau)}{\sum_{j=1}^{N} \exp(s_j^w/\tau)}\right), w \in \{img, batch\}. \tag{12}$$

To ensure the original generalization ability of CLIP, we impose feature-level regularization:

$$\mathcal{L}_{reg} = D\left(f_{img}, f_{img}^{CLIP}\right) + D\left(f_{txt}, f_{txt}^{CLIP}\right) \tag{13}$$

where $D(\cdot, \cdot)$ is the cosine distance, and $f^{CLIP}$ are features from the frozen CLIP encoder.

The final loss can be computed as follows:

$$\mathcal{L}_{total} = \alpha_2 \cdot \mathcal{L}_{CE}^{img} + (1 - \alpha_2) \cdot \mathcal{L}_{CE}^{batch} + \lambda \cdot \mathcal{L}_{reg} \tag{14}$$

**Inference Phrase.** It should be noted that prototypes are used only during training to guide better clustering of representations; they are not used at inference. And our inference strategy differentiates between base and novel classes as illustrated in Figure 3. For base classes, we compute ensemble logits as a weighted sum of image-text and batch-text scores, where both $f_{img}$ and $f_{batch}$ are compared against the text features $f_{txt}$:

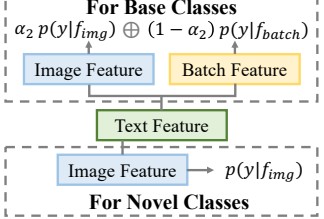

$$p\left(y|f_{img}, f_{batch}\right) = \alpha_2 \cdot p\left(y|f_{img}\right) + (1 - \alpha_2) \cdot p\left(y|f_{batch}\right), \tag{15}$$

For novel classes, we use only image features $f_{img}$ to preserve generalization following Guo & Gu (2025a) (*i.e.* $\alpha_2 = 1$). The final prediction is obtained via $\hat{y} = \arg\max_y p\left(y|f_{img}, f_{batch}\right)$.

Figure 3: Inference on base and novel classes.

## 4 EXPERIMENTS

### 4.1 EXPERIMENTAL SETUP

**Datasets.** We evaluate on 11 standard vision datasets: ImageNet (Deng et al., 2009), Caltech101 (Fei-Fei et al., 2004), OxfordPets (Parkhi et al., 2012), StanfordCars (Krause et al., 2013),

Table 2: Comparison with state-of-the-art methods on base-to-novel generalization across 11 datasets. The best result are in **bold**, and the second best result are underlined.

| | Average | | | ImageNet | | | Caltech101 | | | OxfordPets | | |
|---|---|---|---|---|---|---|---|---|---|---|---|---|
| | Base | Novel | HM | Base | Novel | HM | Base | Novel | HM | Base | Novel | HM |
| CLIP | 69.49 | 74.30 | 71.82 | 72.40 | 68.10 | 70.18 | 97.20 | 94.20 | 95.68 | 91.30 | 97.10 | 94.11 |
| CoOp | 82.23 | 67.94 | 74.41 | 76.33 | 67.73 | 71.77 | 98.23 | 93.10 | 95.60 | 94.47 | 95.57 | 95.02 |
| CoCoOp | 80.63 | 72.47 | 76.33 | 76.00 | 70.57 | 73.18 | 97.73 | 93.30 | 95.46 | 94.93 | 97.80 | 96.34 |
| KgCoOp | 81.18 | 73.45 | 77.12 | 75.87 | 69.83 | 72.72 | 97.83 | 94.47 | 96.12 | 94.88 | 97.60 | 96.22 |
| MaPLe | 81.88 | 74.92 | 78.25 | 76.77 | 70.50 | 73.50 | 97.87 | **95.47** | **96.66** | 95.47 | **98.00** | 96.72 |
| ProVP | 84.70 | 71.81 | 77.72 | 75.88 | 67.93 | 71.69 | 98.77 | 94.21 | 96.44 | 95.04 | 97.11 | 96.06 |
| MMRL | **85.56** | 76.56 | 80.81 | **77.90** | 71.20 | 74.40 | 98.90 | 94.30 | 96.55 | 95.67 | 97.50 | 96.58 |
| MMRL++ | 85.36 | 77.62 | 81.31 | 77.67 | 71.53 | 74.47 | 98.70 | 94.03 | 96.31 | 95.10 | 96.87 | 95.98 |
| ATPrompt | 83.66 | 71.40 | 77.05 | 77.00 | 69.20 | 72.89 | 98.17 | 93.83 | 95.95 | 95.87 | 97.73 | **96.79** |
| **PGMPL** | 85.55 | **78.07** | **81.64** | 77.53 | 71.70 | 74.50 | **98.93** | 94.43 | 96.63 | 95.93 | 97.60 | 96.76 |

| | StanfordCars | | | Flowers102 | | | Food101 | | | FGVCAircraft | | |
|---|---|---|---|---|---|---|---|---|---|---|---|---|
| | Base | Novel | HM | Base | Novel | HM | Base | Novel | HM | Base | Novel | HM |
| CLIP | 63.70 | 74.90 | 68.85 | 71.70 | **77.40** | 74.44 | 90.00 | 91.20 | 90.60 | 27.60 | 35.90 | 31.21 |
| CoOp | 75.63 | 69.37 | 72.36 | 97.60 | 67.43 | 79.76 | 89.20 | 87.47 | 88.33 | 38.10 | 28.00 | 32.28 |
| CoCoOp | 70.80 | 72.43 | 71.61 | 95.03 | 70.90 | 81.21 | 90.57 | 91.13 | 90.85 | 36.00 | 32.53 | 34.18 |
| KgCoOp | 72.72 | 74.78 | 73.74 | 94.87 | 74.59 | 83.52 | 90.47 | 91.70 | 91.08 | 37.30 | 33.57 | 35.34 |
| MaPLe | 72.20 | 74.75 | 73.45 | 95.77 | 74.07 | 83.53 | **90.73** | **92.17** | **91.44** | 36.53 | 35.27 | 35.89 |
| ProVP | 79.40 | 68.67 | 73.65 | 98.07 | 69.88 | 81.61 | 90.29 | 91.05 | 90.67 | 45.60 | 31.29 | 37.11 |
| MMRL | 81.20 | 74.70 | 77.81 | **98.90** | 76.87 | 86.50 | 90.60 | 91.50 | 91.05 | 45.60 | 37.03 | 40.87 |
| MMRL++ | 81.23 | 75.03 | 78.01 | 98.13 | 77.33 | 86.50 | 90.47 | 91.63 | 91.05 | **46.43** | 38.57 | 42.14 |
| ATPrompt | 77.33 | 72.43 | 74.80 | 97.60 | 69.33 | 81.07 | 89.83 | 90.80 | 90.31 | 39.93 | 24.90 | 30.67 |
| **PGMPL** | **82.13** | **75.73** | **78.80** | 98.47 | 76.97 | 86.40 | 89.87 | 91.97 | 90.91 | 46.20 | **38.83** | **42.20** |

| | SUN397 | | | DTD | | | EuroSAT | | | UCF101 | | |
|---|---|---|---|---|---|---|---|---|---|---|---|---|
| | Base | Novel | HM | Base | Novel | HM | Base | Novel | HM | Base | Novel | HM |
| CLIP | 69.40 | 75.60 | 72.37 | 53.20 | 60.70 | 56.70 | 57.00 | 63.80 | 60.21 | 70.90 | 78.40 | 74.46 |
| CoOp | 81.13 | 69.27 | 74.73 | 79.37 | 48.80 | 60.44 | 89.73 | 58.57 | 70.88 | 84.70 | 62.07 | 71.64 |
| CoCoOp | 79.40 | 76.20 | 77.77 | 76.97 | 54.27 | 63.66 | 87.37 | 64.77 | 74.39 | 82.10 | 73.27 | 77.43 |
| KgCoOp | 80.47 | 76.80 | 78.59 | 79.01 | 56.52 | 65.90 | 86.31 | 61.72 | 71.97 | 83.23 | 76.33 | 79.63 |
| MaPLe | 80.90 | 78.00 | 79.42 | 80.00 | 58.30 | 67.45 | 91.27 | 69.70 | 79.04 | 83.20 | 77.93 | 80.48 |
| ProVP | 80.66 | 74.87 | 77.66 | 82.64 | 57.53 | 67.84 | **97.44** | 63.74 | 77.07 | 87.88 | 73.64 | 80.13 |
| MMRL | 82.97 | 79.20 | 81.04 | **85.70** | 74.33 | 74.33 | 95.73 | 75.60 | 84.48 | **88.03** | 78.67 | 83.09 |
| MMRL++ | 82.93 | 79.50 | 81.18 | 85.10 | 66.03 | 74.36 | 95.60 | 83.73 | 89.27 | 87.63 | 79.53 | 83.38 |
| ATPrompt | 81.67 | 76.33 | 78.91 | 81.67 | 51.37 | 63.07 | 95.83 | 64.23 | 76.91 | 85.33 | 75.27 | 79.98 |
| **PGMPL** | 82.67 | **79.67** | 81.14 | 85.37 | 66.33 | 74.66 | 96.87 | **86.27** | **91.26** | 87.03 | 79.30 | 82.99 |

Flowers102 (Nilsback & Zisserman, 2008), Food101 (Bossard et al., 2014), FGVCAircraft (Maji et al., 2013), SUN397 (Xiao et al., 2010), DTD (Cimpoi et al., 2014), EuroSAT (Helber et al., 2019), and UCF101 (Soomro et al., 2012), which includes generic object recognition, fine-grained classification, scene understanding, remote sensing and human action recognition. We alse test the cross-domain effect of our method on ImageNetV2 (Recht et al., 2019), ImageNet-Sketch (Wang et al., 2019), ImageNet-A (Hendrycks et al., 2021b) and ImageNet-R (Hendrycks et al., 2021a).

**Implementation Details.** The implementation details can be found in Appendix A.1.1.

**Evaluation.** For image-text classification, we test base-to-novel, cross-dataset and cross-domain generalization, then record the accuracy on the base/novel classes, and their harmonic mean (HM). We also introduce an image-cluster feature classification as a new evaluation, directly assessing whether the prototype-guided visual representations exhibit improved clustering.

## 4.2 BASE-TO-NOVEL GENERALIZATION

We compare our method with the zero-shot baseline CLIP, as well as prompt learning methods, including CoOp (Zhou et al., 2022c), CoCoOp (Zhou et al., 2022b), KgCoOp (Yao et al., 2023), MaPLe (Khattak et al., 2023), ProVP (Xu et al., 2025), MMRL (Guo & Gu, 2025a), MMRL++ (Guo & Gu, 2025b), and ATPrompt (Li et al., 2024).

Table 3: Comparison with state-of-the-art methods MMRL and MMRL++ on image-cluster feature classification evaluation across 11 datasets.

| | Average | | | ImageNet | | | Caltech101 | | | OxfordPets | | |
|---|---|---|---|---|---|---|---|---|---|---|---|---|
| | Base | Novel | HM | Base | Novel | HM | Base | Novel | HM | Base | Novel | HM |
| CLIP | 65.70 | 68.38 | 67.01 | 51.38 | 49.95 | 50.65 | 94.80 | 89.09 | 91.86 | 67.99 | 69.33 | 68.65 |
| MMRL | 70.64 | 71.60 | 71.12 | 55.31 | 53.14 | 54.20 | 95.00 | 90.96 | 92.94 | 82.15 | 79.06 | 80.58 |
| MMRL++ | 69.92 | 70.53 | 70.22 | 55.26 | 53.20 | 54.21 | 94.84 | 88.99 | 91.82 | 77.95 | 69.13 | 73.28 |
| PGMPL | **72.41** | **73.34** | **72.87** | **58.17** | **55.38** | **56.74** | **95.86** | **92.04** | **93.91** | **84.81** | **80.98** | **82.85** |

| | StanfordCars | | | Flowers102 | | | Food101 | | | FGVCAircraft | | |
|---|---|---|---|---|---|---|---|---|---|---|---|---|
| | Base | Novel | HM | Base | Novel | HM | Base | Novel | HM | Base | Novel | HM |
| CLIP | 50.33 | 62.51 | 55.76 | 89.22 | 88.99 | 89.10 | 74.02 | 78.07 | 75.99 | 28.57 | 37.64 | 32.48 |
| MMRL | 55.22 | 65.02 | 59.72 | 92.20 | 90.90 | 91.55 | 75.32 | 78.87 | 77.05 | 31.68 | 41.40 | 35.89 |
| MMRL++ | 55.03 | 64.50 | 59.39 | 90.68 | 91.12 | 90.90 | 74.43 | 78.01 | 76.18 | 32.54 | 42.64 | 36.91 |
| PGMPL | **58.18** | **68.04** | **62.72** | **93.20** | **91.89** | **92.54** | **77.36** | **81.30** | **79.28** | **33.47** | **42.71** | **37.53** |

| | SUN397 | | | DTD | | | EuroSAT | | | UCF101 | | |
|---|---|---|---|---|---|---|---|---|---|---|---|---|
| | Base | Novel | HM | Base | Novel | HM | Base | Novel | HM | Base | Novel | HM |
| CLIP | 60.87 | 64.47 | 62.62 | 56.55 | 59.21 | 57.85 | 82.47 | 86.07 | 84.23 | 66.49 | 66.85 | 66.67 |
| MMRL | 63.81 | 66.63 | 65.19 | 64.35 | 62.45 | 63.39 | 94.02 | 91.34 | 92.66 | 67.99 | 67.85 | 67.92 |
| MMRL++ | 63.70 | 66.81 | 65.22 | 62.57 | 61.56 | 62.06 | 93.98 | **91.82** | 92.89 | 68.10 | 68.00 | 68.05 |
| PGMPL | **66.64** | **69.50** | **68.04** | **65.68** | **64.73** | **65.20** | **94.13** | 90.72 | 92.39 | **69.02** | **69.45** | **69.23** |

Table 2 compares the base-to-novel generalization performance of our PGMPL with other state-of-the-art methods across 11 datasets. On average, our method achieves an HM of 81.64%, surpassing the state-of-the-art method MMRL++ by 0.33%. Specifically, PGMPL improves accuracy on novel classes by 0.45% over MMRL++, and is 0.19% higher on base classes, remaining comparable to the best MMRL. These results indicate that PGMPL effectively enhances adaptation to unseen classes while maintaining high accuracy on base classes. Beyond the average metrics, PGMPL's effectiveness is validated on multiple individual datasets. On 9 datasets, our approach either outperforms or is comparable to the current state-of-the-art methods, demonstrating its effectiveness and universality.

### 4.3 IMAGE-CLUSTER FEATURE CLASSIFICATION EVALUATION

We further aggregate the features of five given images to form a cluster feature, which is then matched against images, and compare the results with CLIP and state-of-the-art methods to verify that prototype guidance leads the model to learn better visual representations. As shown in Table 3, our method achieves the best accuracy on 10 datasets, with the average HM exceeding the best prior method by 1.75%; accuracy improves by 1.77% on base classes and by 1.74% on novel classes. In this setting, PGMPL shows a larger advantage over state-of-the-art methods, indicating that prototype guidance enhances visual representations; in turn, the improved visual features further boost image-text matching performance, as shown in Table 2. We also use t-SNE plots to show the distribution of visual representations for some datasets in Appendix A.3.

We find that accuracy is limited on certain datasets in the base-to-novel generalization. For instance, on Flowers101, image-text matching achieves only 76.97% on novel class in Table 2, whereas image-cluster feature matching raises it to 91.89% (+14.92%) in Table 3. This indicates that text-based matching can be a performance bottleneck in some scenarios, while class-level visual cluster features can capture fine-grained semantics that text prompts struggle to express, thereby providing more discriminative representations. These observations also demonstrate the necessity of introducing a class-level summarizing prototype as a supervisory signal to improve model's generalization.

### 4.4 CROSS-DATASET AND CROSS-DOMAIN EVALUATION

Table 4 shows the performance of models trained on ImageNet and transferred to other datasets, covering both cross-dataset and cross-domain settings. Our method achieves an average accuracy of 65.41%, comparable to the state-of-the-art method MMRL. However, in the base-to-novel generalization experiment, MMRL's HM is lower than ours by 0.83% (see Table 2). Moreover, the previous best method, MMRL++, is lower than ours by 0.09% on the cross-dataset and cross-domain evalu-

Table 4: Comparison with state-of-the-art methods on cross-dataset and cross-domain evaluation across 11 datasets.

| | Source | Target | | | | | | | | | | | | | | |
| | ImNet | Caltech | Pets | Cars | Flowers | Food | Aircraft | SUN397 | DTD | EuroSAT | UCF101 | ImNetV2 | ImNet-S | ImNet-A | ImNet-R | *Average* |
|---|---|---|---|---|---|---|---|---|---|---|---|---|---|---|---|---|
| CLIP | 66.70 | 92.90 | 89.10 | 65.30 | 71.30 | 86.10 | 24.80 | 62.60 | 44.50 | 47.50 | 66.80 | 60.80 | 46.10 | 47.80 | 74.00 | 62.83 |
| CoOp | 71.50 | 93.43 | 89.10 | 63.43 | 69.40 | 85.37 | 18.13 | 64.47 | 41.10 | 41.40 | 66.67 | 64.13 | 48.17 | 50.23 | 76.03 | 62.22 |
| CoCoOp | 71.13 | 94.47 | 90.60 | 65.33 | 71.57 | 86.10 | 22.63 | 67.17 | 45.23 | 46.93 | 68.73 | 64.33 | 48.87 | **50.97** | 76.53 | 64.25 |
| KgCoOp | 70.60 | 93.67 | 90.00 | 65.63 | 70.33 | 86.40 | 23.13 | 66.37 | 46.43 | 43.43 | 68.27 | 63.83 | 48.57 | 50.47 | 76.70 | 63.80 |
| MaPLe | 70.60 | 93.93 | 90.90 | 65.47 | 71.23 | 86.13 | 23.47 | 67.10 | 45.60 | 46.90 | 67.40 | 63.97 | 48.77 | 50.83 | 76.93 | 64.19 |
| ProVP | **75.88** | 92.51 | 89.11 | 61.81 | 64.55 | 82.74 | 23.24 | 63.58 | 43.62 | 41.70 | 66.05 | 61.26 | 45.29 | 43.50 | 72.92 | 60.85 |
| MMRL | 72.03 | 94.53 | 91.67 | 66.03 | 72.77 | 86.40 | 26.23 | 67.43 | 46.43 | 53.10 | 68.77 | **64.67** | 49.17 | 50.93 | 77.60 | **65.41** |
| MMRL++ | 71.87 | 94.57 | 91.37 | 66.33 | **73.20** | **86.70** | 25.90 | **67.67** | 45.80 | 51.90 | **69.00** | 64.40 | **49.20** | 50.87 | **77.63** | 65.32 |
| ATPrompt | 70.80 | 93.97 | 89.90 | 63.00 | 69.40 | 86.07 | 22.43 | 65.23 | 42.23 | 46.13 | 65.67 | 64.20 | 48.10 | 50.57 | 76.50 | 63.10 |
| **PGMPL** | 71.13 | **94.80** | **91.70** | **66.37** | 72.37 | 86.50 | **26.40** | 67.60 | **46.57** | **53.87** | 68.50 | 64.60 | 48.87 | 50.17 | 77.37 | **65.41** |

ation. Across the 14 datasets, we obtain state-of-the-art results on 6 and second-best on 3, demonstrating stable transferability and achieving the best average performance across different domains.

## 4.5 ABLATION STUDY

**Module ablations.** We conduct ablations on different modules in Table 5 to verify the contribution of each component. Specifically, "w/o batch token" removes in-batch aggregation and retains only basic learnable tokens for prompt learning; "w/o cross-attention" inserts learnable tokens into the image and text branches independently without cross-modal interaction; "w/o prototype" trains with the standard image-text contrastive loss only.

Table 5: Ablation on different modules.

| Variants | Base | Novel | HM |
|---|---|---|---|
| w/o cross-attention | **85.70** | 76.67 | 80.93 |
| w/o batch tokens | 85.41 | 76.60 | 80.77 |
| w/o prototype | 85.23 | 76.27 | 80.51 |
| PGMPL | 85.55 | **78.07** | **81.64** |

We observe that adding cross-attention slightly reduces performance on base classes but substantially improves novel performance by 1.40%, indicating that cross-attention effectively enhances feature generalization. We also compare with other cross-modal interaction strategies in Appendix A.2.1, further demonstrating the effectiveness of our bidirectional cross-attention interaction mechanism. In addition, our design of batch token not only adds learnable tokens but also aggregates in-batch information, effectively improving the quality of representation. Finally, introducing the prototype further boosts performance on both base and novel classes, suggesting that a modality-bridging and category-level supervisory signal strengthens generalization and discriminability of representations.

**More ablations.** Additional ablations on learnable tokens interaction strategies, the number of batch tokens, and different parameters are provided in Appendix A.2.3.

## 5 CONCLUSION

We propose a novel prompt learning method PGMPL, which improves the generalization ability of VLMs in zero-shot scenarios. We introduce a modality-bridging and category-level prototype to guide representation learning, aiming to enhance CLIP's generalization to unseen classes. To improve representation quality, we insert batch tokens into intermediate encoder layers, and employ bidirectional cross-attention to mitigate the image-text misalignment caused by the inserted batch tokens. We then update per-class prototypes based on the learned feature and use them as supervision and guidance to further optimize image-text matching. Extensive experiments demonstrate that our method surpasses current state-of-the-art approaches on base-to-novel image-text and image-cluster feature matching tasks, and achieves comparable results in cross-dataset and cross-domain settings, showcasing its potential for zero-shot learning applications.

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

# A APPENDIX

## A.1 EXPERIMENT SETUP

### A.1.1 MORE IMPLEMENTATION DETAILS

We utilize a frozen ViT-B/16 CLIP backbone (Radford et al., 2021), training only learnable tokens and auxiliary modules as shown in Figure 1. All experiments follow a 16-shots setting, where 16 images per base class are sampled for training.

We employ the AdamW optimizer with a learning rate of $10^{-3}$. The number of batch token is $M = 10$, and we insert learnable tokens starting from the sixth layer of the encoder ($K = 6$). Regarding the design of batch token, we set different batch size $Z$ and aggregation weights $\beta_1$ and $\beta_2$ for batch tokens across datasets, as shown in Appendix A.2.3. Then, we set the shallow-deep encoder information fusion weight $\beta_3$ to 0.5. For loss weights, the prototype guidance coefficient $\alpha_1$ is set to 0.7 on ImageNet and 0.5 for other datasets, while the cross-entropy term weight $\alpha_2 = 0.7$. The regularization weight $\lambda$ is also set per dataset following MMRL (Guo & Gu, 2025a). Finally, the momentum update coefficient for prototypes is $\gamma = 0.9$.

For the base to novel generalization and image-cluster feature classification evaluation experiments, the epoch for ImageNet is set to 5, and for other datasets, it is set to 10. For other experiments, the epoch is set to 1. All methods are trained and tested under identical conditions, with metrics averaged over three independent trials on a single NVIDIA L20 GPU.

## A.2 EXPERIMENTS

### A.2.1 DETAILED RESULTS OF ABLATION ANALYSIS ON LEARNABLE TOKENS INTERACTION

We compare different cross-modal interaction strategies in Table 6. "Text to image" maps tokens unidirectionally from the text embedding to the image embedding; "shared space" first defines shared tokens and then maps them to the image and text embedding separately; "cross-attention" uses bidirectional cross-attention for direct cross-modal interaction, corresponding to Figure 2 (b)-(d).

Table 6: Ablation study on interaction strategies across 11 datasets.

| Interaction strategies | Base | Novel | HM |
|---|---|---|---|
| text to image | 85.63 | 75.89 | 80.46 |
| shared space | **85.67** | 76.47 | 80.80 |
| cross-attention (PGMPL) | 85.55 | **78.07** | **81.64** |

Using text-to-image mapping directly for cross-modal interaction can introduce bias, thereby weakening generalization to novel classes. Although the shared space approach achieves the best performance on base classes, this indirect interaction by projecting learnable tokens to the two branches separately still lacks sufficient generalization, resulting in poorer performance on novel classes. In contrast, the bidirectional cross-attention mechanism enables more direct cross-modal interaction, effectively alleviates image-text misalignment, and captures finer-grained semantic correspondences, thereby substantially improving zero-shot generalization.

### A.2.2 DETAILED RESULTS ON ALL 11 DATASETS OF ABLATION ANALYSIS ON THE NUMBER OF BATCH TOKENS

We conduct an ablation on the number of learnable tokens, varying it in $\{4, 6, 8, 10, 12, 14\}$, and compare our method with the current state-of-the-art method MMRL++, in terms of base, novel, and harmonic-mean (HM) accuracy, as shown in Figure 4.

For our PGMPL, as the number of tokens increases from 4 to 10, both novel and HM steadily improve and peak at 10 tokens (Novel $= 78.07\%$, HM $= 81.64\%$). When tokens $> 10$, accuracy decreases slightly but remains clearly higher than MMRL++. For MMRL++, the curves are almost flat when tokens $\leq 10$, indicating little benefit from adding more tokens; when tokens $> 10$, performance degrades. For example, at 12 tokens, the novel accuracy of MMRL++ is 1.31% lower than ours. Overall, the two methods are similar on base classes, while our gains mainly lie in generalization to novel classes, resulting in a higher HM and validating the effectiveness of our approach.

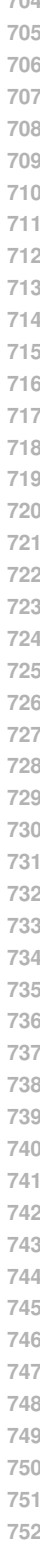

Figure 4: Ablation results on the number of batch tokens.

Table 7: Ablation study on batch size across 11 datasets. All results are reported in terms of HM.

| $Z$ | ImNet | Caltech | Pets | Cars | Flowers | Food | Aircraft | SUN397 | DTD | EuroSAT | UCF |
|---|---|---|---|---|---|---|---|---|---|---|---|
| 4 | **74.41** | **96.52** | 96.17 | **78.71** | 86.27 | 90.77 | **41.85** | **81.05** | 74.19 | 87.67 | 82.93 |
| 8 | 74.32 | 96.37 | 96.20 | 78.42 | **86.34** | 90.84 | 41.63 | 80.99 | **74.66** | **90.87** | 82.91 |
| 16 | 74.41 | 96.22 | 96.58 | 78.46 | 86.03 | **90.87** | 41.69 | 80.92 | 73.87 | 88.53 | **82.93** |
| 32 | 74.29 | 96.47 | **96.64** | 78.07 | 86.21 | 90.79 | 39.65 | 80.94 | 73.48 | 84.40 | 81.94 |

Table 8: Ablation study on $\beta_1$ and $\beta_2$ across 11 datasets.

| $\beta_1$ | $\beta_2$ | ImNet | Caltech | Pets | Cars | Flowers | Food | Aircraft | SUN397 | DTD | EuroSAT | UCF |
|---|---|---|---|---|---|---|---|---|---|---|---|---|
| 0.0 | 0.0 | 74.33 | 96.37 | 96.48 | 78.63 | 86.12 | 90.85 | 41.89 | 81.03 | 74.39 | 88.91 | 82.76 |
| 0.4 | 0.4 | 74.40 | 96.56 | 96.66 | 78.67 | 86.35 | 90.90 | 41.81 | 81.15 | 74.57 | 91.13 | 82.94 |
| 0.4 | 0.5 | 74.38 | 96.56 | 96.71 | 78.55 | 86.34 | 90.89 | 41.99 | 81.06 | 74.66 | **91.26** | 82.85 |
| 0.4 | 0.6 | 74.38 | **96.63** | 96.73 | 78.67 | 86.32 | 90.90 | 41.88 | 81.11 | 74.43 | 91.17 | 82.86 |
| 0.5 | 0.4 | 74.42 | 96.53 | 96.51 | 78.78 | **86.40** | 90.87 | 42.05 | **81.14** | 74.62 | 90.56 | 82.99 |
| 0.5 | 0.5 | 74.41 | 96.52 | 96.64 | 78.71 | 86.34 | 90.87 | 41.85 | 81.05 | **74.66** | 90.87 | 82.93 |
| 0.5 | 0.6 | 74.36 | 96.59 | **96.76** | 78.69 | 86.29 | 90.89 | **42.20** | 81.09 | 74.36 | 90.83 | 82.97 |
| 0.6 | 0.4 | 74.38 | 96.48 | 96.45 | 78.76 | 86.28 | 90.87 | 41.86 | 81.11 | 74.50 | 89.60 | **82.99** |
| 0.6 | 0.5 | **74.45** | 96.46 | 96.55 | **78.80** | 86.27 | 90.87 | 41.89 | 81.12 | 74.66 | 89.40 | 82.97 |
| 0.6 | 0.6 | 74.43 | 96.46 | 96.61 | 78.79 | 86.32 | **90.91** | 42.01 | 81.02 | 74.22 | 89.40 | 83.01 |
| 1.0 | 1.0 | 74.31 | 96.51 | 96.40 | 78.73 | 85.68 | 90.82 | 42.05 | 81.12 | 73.65 | 83.74 | 82.82 |

### A.2.3 DETAILED RESULTS ON ALL 11 DATASETS OF ABLATION ANALYSIS ON DIFFERENT PARAMETERS

We initialize the batch size $Z$ for each dataset to 8, set $\beta_1 = \beta_2 = \beta_3 = 0.5$, $\alpha_1 = 0.5$, and $\alpha_2 = 0.7$, and then perform ablation studies on these parameters.

**Ablation analysis on batch size $Z$.** Since token computation is batch-dependent, and larger batch sizes incorporate more intra-batch information during token optimization, we conduct an ablation study on batch size $Z$. To analyze how this impacts performance across datasets, we evaluate batch sizes of 4, 8, 16, and 32. As shown in Table 7, the optimal batch size varies per dataset, reflecting differences in their utilization of batch-level information.

**Ablation analysis on $\beta_1$ and $\beta_2$.** We next perform ablation studies on $\beta_1$ and $\beta_2$, which govern the fusion of randomly initialized tokens with intra-batch information in both vision and text modalities. Specifically, $\beta_1$ controls the weight of batch-level features in image-side token optimization, while $\beta_2$ controls the same for text-side. Due to modality differences, the optimal parameters can differ across modalities. To

Table 9: Ablation study on $\beta_3$ across 11 datasets.

| $\beta_3$ | Base | Novel | HM |
|---|---|---|---|
| 0.0 | 85.40 | 77.38 | 81.20 |
| 0.3 | 85.46 | 77.57 | 81.32 |
| 0.5 | **85.54** | **78.07** | **81.63** |
| 0.7 | 85.48 | 77.43 | 81.26 |
| 1.0 | 85.47 | 76.70 | 80.85 |

Table 10: Ablation study on $\alpha_1$ across 11 datasets.

(a) Average Performance on ImageNet.

| $\alpha_1$ | Base | Novel | HM |
|------|------|-------|------|
| 0.0 | 30.27 | 41.93 | 35.16 |
| 0.3 | 77.43 | 70.73 | 73.93 |
| 0.5 | 77.50 | 71.63 | 74.45 |
| 0.7 | **77.53** | **71.70** | **74.50** |
| 1.0 | 77.47 | 71.63 | 74.44 |

(b) Average Performance across 11 datasets.

| $\alpha_1$ | Base | Novel | HM |
|------|------|-------|------|
| 0.0 | 27.57 | 33.52 | 30.26 |
| 0.3 | **85.60** | 77.40 | 81.29 |
| 0.5 | 85.55 | **78.07** | **81.64** |
| 0.7 | 85.39 | 77.51 | 81.26 |
| 1.0 | 85.24 | 76.28 | 80.51 |

systematically explore the relationship between them, we conduct joint ablation studies within the range $[0.4, 0.6]$ and select the best-performing configurations for each dataset. The results are summarized in Table 8.

**Ablation analysis on $\beta_3$.** We further conduct ablation studies on $\beta_3$, which controls the integration of information between the shallow and deep layers of the encoder. Lower $\beta_3$ indicates greater utilization of the generalized features of shallow layers. By evaluating the $\beta_3$ settings of 0.0, 0.3, 0.5, 0.7, and 1.0, we identify the optimal balance between shallow and deep token information. As shown in Table 9, the best performance is achieved at $\beta_3 = 0.5$, demonstrating that a synergistic combination of both shallow generalization and deep specialization is critical to balance the performance of model on base and novel classes.

**Ablation analysis on $\alpha_1$.** We conduct ablation studies on $\alpha_1$, the weighting coefficient for prototype guidance during training. Lower $\alpha_1$ values correspond to stronger prototype influence. For ImageNet, ablation results (Table 10 (a)) show that $\alpha_1 = 0.7$ achieves the best generalization performance. We then fix $\alpha_1 = 0.7$ for ImageNet and perform ablation studies on other datasets (Table 10 (b)), ultimately adopting $\alpha_1 = 0.5$ for other datasets.

**Ablation analysis on $\alpha_2$.** Ablation analysis on $\alpha_2$. We perform ablation studies on $\alpha_2$, which controls the dependency ratio between image features and batch token features. Lower $\alpha_2$ values indicate higher emphasis on batch token features. As shown in Table 11, we evaluate $\alpha_2$ values of 0.0, 0.3, 0.5, 0.7, and 1.0, and select $\alpha_2 = 0.7$ as the final parameter configuration.

A.3 VISUALIZATION

We visualize the t-SNE distributions of image embedding on EuroSAT, Caltech101, OxfordPets, and Flowers101 datasets, comparing against the state-of-the-art MMRL++ method. As shown in Figure 5, our approach demonstrates better intra-class compactness and higher inter-class separability in different image recognition tasks, indicating that prototype-guided representation learning produces more discriminative features.

Table 11: Ablation study on $\alpha_2$ across 11 datasets.

| $\alpha_2$ | Base | Novel | HM |
|------|------|-------|------|
| 0.0 | 83.33 | 74.03 | 78.40 |
| 0.3 | 84.03 | 76.75 | 80.23 |
| 0.5 | 84.74 | 77.54 | 80.98 |
| 0.7 | **85.55** | **78.07** | **81.64** |
| 1.0 | 83.52 | 76.25 | 79.72 |

A.4 LLM USAGE STATEMENT

During the preparation of this manuscript, we utilized LLM as a writing assistance tool for language polishing. All suggestions from the LLM were critically reviewed, edited, and revised by the authors to ensure the final text accurately reflects our research. The authors take full responsibility for all content presented in this paper.

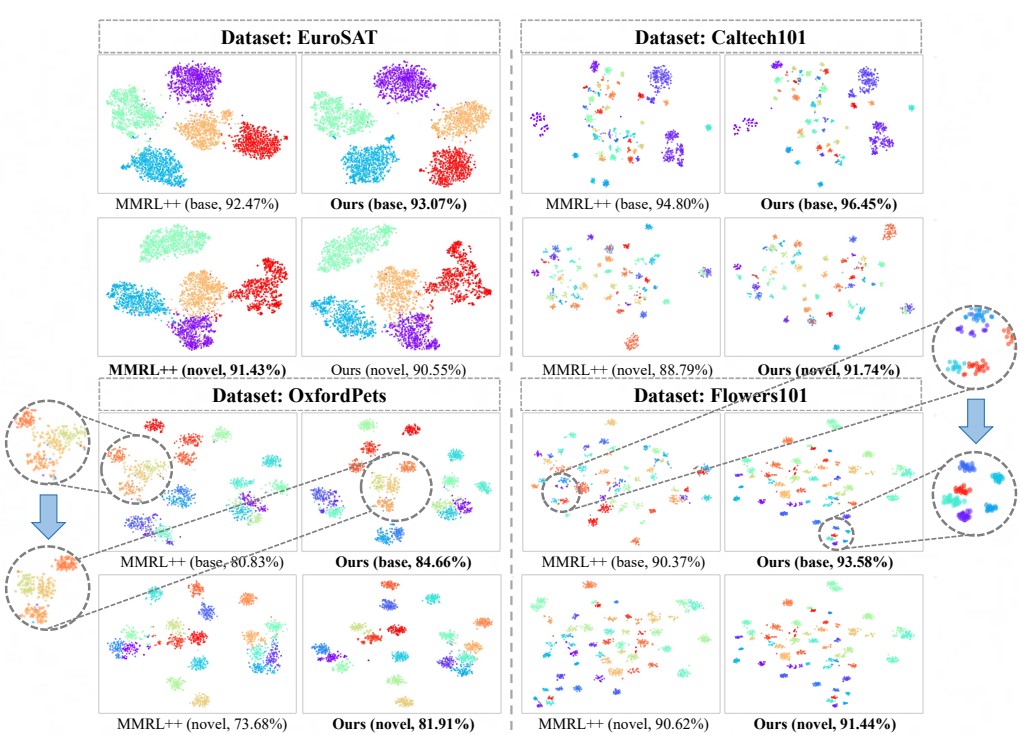

Figure 5: The t-SNE distributions of image embedding on EuroSAT, Caltech101, OxfordPets, and Flowers101 datasets.

