# OpenReview forum: "PGMPL: Prototype-Guided Multi-modal Prompt Learning for Vision-Language Models"
_ICLR.cc/2026/Conference — ICLR 2026 Conference Withdrawn Submission_

### Official Review · Reviewer_9F69 · 2025-10-25

**Soundness:** 2
**Presentation:** 3
**Contribution:** 2
**Rating:** 2
**Confidence:** 4

**Summary:**

The paper introduces the Prototype-Guided Multi-Modal Prompt Learning (PGMPL) framework toenhance the generalization of vision-language models. The core idea is to use category-level prototypes by aggregating features from multiple images and their corresponding texts; these prototypes serve as cross-modal supervision that drives more discriminative representation learning. he method also introduces bidirectional cross-attention and batch tokens to align image and text embeddings while maintaining generalization.

**Strengths:**

1. The overall idea is intuitive and straightforward.

2. The experimental results are comprehensive, with 11 datasets and different task settings.

3. The code is attached, making the method reproducible.

**Weaknesses:**

1. The proposed method offers only incremental innovation, as most modules closely follow existing designs. The prototype-guided learning concept has already been explored in prior works, and the bidirectional cross-attention is highly similar to the mechanism used in MMRL, differing mainly in implementation rather than in principle. The overall framework, both in training and inference, appears too close to MMRL [1] and MMRL++[2], with limited originality across its components.

2. No complexity or efficiency analysis (computational cost, token overhead, training time) is reported, which is critical for prompt-based adaptation methods.

3. It is unclear how prototypes are initialized.

4. The overall flow could be improved by introducing a small algorithm box or pseudo-code to clarify the pipeline.

5. The method involves a large number of coefficients (α₁, α₂, β₁–β₃, γ, λ), and their optimal settings appear to differ across datasets, requiring manual tuning for each case. This per-dataset hyperparameter sensitivity reduces the method’s practical scalability and robustness, making it less convenient for real-world deployment or unseen-domain adaptation.

[1] MMRL: Multi-Modal Representation Learning for Vision-Language Models.  Yuncheng Gu et al.,CVPR2025
[2] MMRL++: Parameter-Efficient and Interaction-Aware Representation Learning for Vision-Language Models. Yuncheng Gu et al.

**Questions:**

1. For datasets such as Flowers102, Food101, and UCF101 in Table 2, the improvements appear marginal. What could be the possible reasons for this limited gain? Additionally, in Table 2 and Table 3, the performance comparison with MMRL++ on EuroSAT shows opposite trends—could the authors clarify why PGMPL performs worse in Table 2 but better in Table 3 under different evaluation settings?

2. The paper lacks sufficient justification and ablation analysis regarding the motivation, effectiveness, and possible alternatives of dividing the batch tokens into two groups (M/2 and M). Could the authors provide a more comprehensive analysis or discussion on this design choice?

3. Although the authors state that learnable tokens are inserted starting from the sixth layer of the encoder, which is a reasonable choice, could they provide ablation results for different insertion layers (e.g., the 4th, 6th, and 8th layers) to make the design choice more convincing?

---

### Official Review · Reviewer_VUDf · 2025-10-30

**Soundness:** 3
**Presentation:** 3
**Contribution:** 2
**Rating:** 4
**Confidence:** 4

**Summary:**

The paper aims to enhance the generalization capabilities of vision-language models (VLMs) like CLIP in zero-shot and few-shot tasks, focusing on overfitting when fine-tuned on downstream tasks. To address this, PGMPL introduces a category-level prototype as a supervisory signal, constructed by aggregating multi-image features with textual semantics using cross-attention. Extensive experiments are conducted on 11 datasets under settings like base-to-novel generalization, cross-dataset, cross-domain, and image-cluster feature matching. Results show PGMPL outperforms state-of-the-art methods, with improvements in novel-class accuracy by 0.45%, overall average by 0.33%, and harmonic mean.

**Strengths:**

1. The prototype mechanism effectively incorporates category-level summaries, leading to more discriminative and aligned representations.
2. Bidirectional cross-attention provides better alignment than unidirectional or shared-space methods, preserving modality-specific information.

**Weaknesses:**

1. The performance boost is not obvious. Compared to previous methods in Table 2 and 3.
2. The motivation is limited. Currently, the generalization problem of VLM has been almost solved by the autoregressive-like models. I doubt the meaning of researching the generalization of CLIP-like models, where the scope is limited.

**Questions:**

See the weakness

---

### Official Review · Reviewer_JHN2 · 2025-11-01

**Soundness:** 2
**Presentation:** 2
**Contribution:** 2
**Rating:** 4
**Confidence:** 4

**Summary:**

This paper proposes the prototype-guided multi-modal prompt learning (PGMPL), a novel framework to help pretrained VLMs (e.g., CLIP) to learn optimal prompt tokens in the embedding space for downstream tasks. One of the core ideas of PGMPL is to design a category-level prototype for each class to summarize the class-level features. PGMPL also develops the batch tokens to enable alignments between the visual and textual encoders. Extensive experiments show the effectiveness of the proposed model.

**Strengths:**

1), The prototype-aware strategy is interesting, and the results show also demonstrate the improvements of the proposed model.

2), The introduced batch tokens is new for the prompt tuning community, which may provide valuable insights for this community.

3), The cross-attention strategy between the image and text tokens help to align the domain gap between the visual and texutal modalities.

**Weaknesses:**

1), Although the idea of batch tokens is interesting, I find it difficult to understand the main motivation behind their introduction. What are the differences between the batch tokens and the prompt tokens used in previous works (e.g., CoOp, MaPLe)? Are these batch tokens shared across all classes, or are they learned separately for each class? Additionally, why are they referred to as “batch tokens”?

2), Why do the batch tokens need to be divided into two parts?

3), What is the motivation behind the proposed L_{CE}^{batch} and L_{reg}​? The authors should provide a more detailed discussion of these losses.

4), It appears that PGMPL employs prototype features to replace the textual features used in previous prompt-tuning methods (e.g., CoOp and MaPLe). What are the main differences between the prototype features and the textual features? The prototype features are obtained through cross-attention between textual features and visual patches; however, this seems more like a network design choice rather than a core contribution of the proposed prototype-aware strategy.

**Questions:**

1), There are typos in lines 264–271. C_l represents visual tokens, not text class tokens.

2), What are the results on few-shot classification?

3), More related works should be discussed, particularly [1,2,3].

4), A comparison with AWT [3] should be included to better demonstrate the effectiveness of the proposed model, as AWT also enhances CLIP’s transferability to downstream tasks.


[1] Patch-Prompt Aligned Bayesian Prompt Tuning for Vision-Language Models

[2] Tuning Multi-mode Token-level Prompt Alignment across Modalities

[3] AWT: Transferring Vision-Language Models via Augmentation, Weighting, and Transportation

---

### Note · Authors · 2025-11-29

I have read and agree with the venue's withdrawal policy on behalf of myself and my co-authors.